# CROSS-LINGUAL TRANSFER WITH CLASS-WEIGHTED LANGUAGE-INVARIANT REPRESENTATIONS

**Ruicheng Xian, Heng Ji & Han Zhao**
Department of Computer Science
University of Illinois Urbana-Champaign
Urbana, IL 61801, USA
`{rxian2,hengji,hanzhao}@illinois.edu`

## ABSTRACT

Recent advances in neural modeling have produced deep multilingual language models capable of extracting cross-lingual knowledge from non-parallel texts and enabling zero-shot downstream transfer. While their success is often attributed to shared representations, quantitative analyses are limited. Towards a better understanding, through empirical analyses, we show that the invariance of feature representations across languages—an effect of shared representations—strongly correlates with transfer performance. We also observe that distributional shifts in class priors between source and target language task data negatively affect performance, a largely overlooked issue that could cause negative transfer with existing unsupervised approaches. Based on these findings, we propose and evaluate a method for unsupervised transfer, called importance-weighted domain alignment (IWDA), that performs representation alignment with prior shift estimation and correction using unlabeled target language task data. Experiments demonstrate its superiority under large prior shifts, and show further performance gains when combined with existing semi-supervised learning techniques.

## 1 INTRODUCTION

Many recent state-of-the-art results on natural language processing (NLP) tasks are achieved on transformer-based deep neural language models (LMs) under the "pre-train then fine-tune" paradigm (Devlin et al., 2019; Conneau & Lample, 2019). Multilingual versions of these LMs are pre-trained on unannotated and non-parallel texts in more than one language, such as multilingual BERT (mBERT) and XLM-R (Conneau et al., 2020a), and their transformer encoder is shared across languages. The cross-lingual knowledge these models acquired has enabled *zero-shot* transfer, meaning that after fine-tuned for a downstream task in one source language, not only would they work well when evaluated in the source, but also quite decently in almost all languages seen during pre-training (Wu & Dredze, 2019).

Their success on zero-shot cross-lingual transfer has prompted numerous studies on their multilingual abilities. One recurring hypothesis is that the deep architecture of these models combined with parameter sharing induced intermediate feature representations that are shared across languages (Karthikeyan et al., 2020; Conneau et al., 2020b; Muller et al., 2021), but they fall short of providing insights to the question of *how shared representations contribute to cross-lingual learning?*

A potential effect of shared representations is *representation invariance*, where the model outputs similar feature representations for semantically similar inputs across all languages (Zhao et al., 2020). Having language-invariant features means that any predictor trained in one language can be immediately transferred to other languages, giving the same predictions on inputs with similar semantics. For classification tasks, the weaker condition of *class-conditional invariance* is sufficient for unsupervised cross-lingual transfer, where the distributions of the features conditioned on each class label are aligned across languages (Zhao et al., 2019; Ben-David et al., 2007).

---

Our code is available at `https://github.com/rxian/domain-alignment`.

We hypothesize that deep transformer-based multilingual LMs achieve cross-lingual transfer with class-conditional language-invariant representations acquired during pre-training and fine-tuning. To provide quantitative evidence, through empirical analyses on mBERT and XLM-R, we show that (1) transfer performance strongly correlates with the class-conditional alignment of feature representations for the downstream task (Section 2.1). During our studies, we also observed that (2) performance is negatively affected by distributional shifts in the class priors between source and target task data (Section 2.2). Despite being a common issue with real-world data, prior shifts have been largely overlooked, and existing unsupervised cross-lingual transfer methods could cause negative transfer under its presence (Section 4.2).

Based on the above finding, for cross-lingual learning under the unsupervised setting where access to target language unlabeled task data is available along with source language labeled data, we propose importance-weighted domain alignment (IWDA). This method aims to learn language-invariant representations with feature alignment while accounting for the class prior shift with an importance weight estimation and correction procedure (Section 3). Our experiments and evaluations on multilingual sentiment analysis, named-entity recognition, and textual entailment show its effectiveness for unsupervised transfer and superiority over the more common semi-supervised learning (SSL) methods under large prior shifts. We also show that IWDA and SSL are compatible, where they can be combined to achieve further performance gains. The results are detailed in Section 4, along with analyses and discussions.

## 2 Two Factors Affecting Cross-Lingual Transfer

This section describes the experiment setup for the empirical analyses on the effects of representation invariance and class prior shift on cross-lingual transfer performance, and discusses their implications on the design of unsupervised transfer methods.

**Background.** We introduce notation by briefly summarizing the "pre-train then fine-tune" paradigm below. Let $g : \mathcal{X} \to \mathcal{Z}$ denote the feature mapping provided by the pre-trained LM ($\mathcal{Z} \subset \mathbb{R}^{768}$ in BERT-Base), and $z = g(x)$ the feature computed on an input token or sentence $x$. For token classification tasks, the transformer's last-layer contextualized token embeddings are taken as features. For sequence classification, we follow the practice of using the embedding of the start-of-sentence marker (symbolized as `[CLS]`); alternatives include mean-pooling the token embeddings. To compute class label predictions $\hat{y}$ from the features, a single linear layer with softmax activation $h : \mathcal{Z} \to \mathcal{Y} = \{1, 2, \cdots, k\}$ is added on top of $g$, so that $\hat{y} = h(g(x))$.

This end-to-end model is transferred and adapted to the downstream task by simultaneously fine-tuning $g$ and training $h$ on task data. In zero-shot settings, fine-tuning is performed on source language labeled data only, $(x, y) \sim p_S$, without access to any target language task data $p_T$.

**Setup.** Our analysis begins with the following decomposition of the joint feature-label distribution into a product of class-conditional feature distribution and marginal prior distribution,

$$p(z, y) = p(z|y)p(y) := p(\{x \in \mathcal{X} : g(x) = z\}|Y = y)p(Y = y).$$

Achieving representation invariance between the source and target language means that $p_S(z|y) = p_T(z|y)$ for all $z \in \mathcal{Z}$, $y \in \mathcal{Y}$, and the presence of *class prior shift* means that $p_S(y) \neq p_T(y)$ for some $y$. For simplicity, we use the shorthands $p^{Z|Y}$ and $p^Y$ to denote class-conditional feature distribution and marginal class prior distribution, respectively.

To study their effects on transfer performance empirically, we compare model performance of mBERT (cased) and XLM-R Large against the alignment of their class-conditioned features and prior shift of the dataset on three multilingual downstream classification tasks: sentiment analysis on the Multilingual Amazon Reviews Corpus (MARC) which covers six high-resource languages, named-entity recognition on the WikiANN dataset which covers 39 languages of varying linguistic properties and resources, and textual entailment on the XNLI dataset which covers 15 languages. Unless otherwise noted, models are only fine-tuned on English data. Due to space constraints, we present results from mBERT on MARC in this section while deferring the remaining results to Appendix A.1 and B.

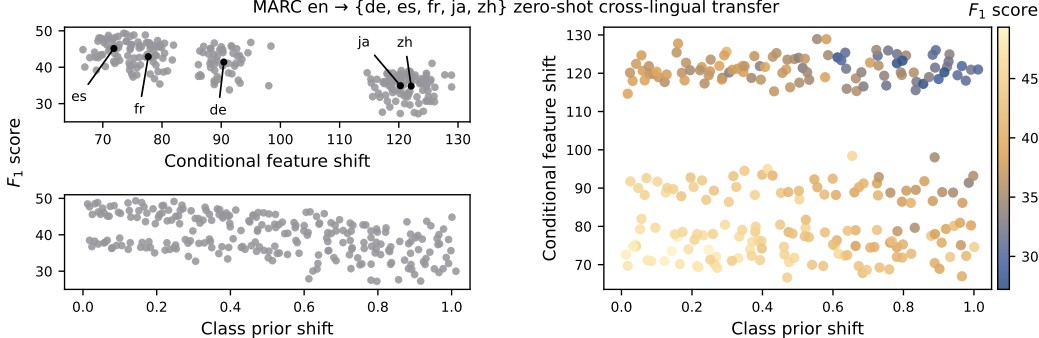

Figure 1: Zero-shot transfer performance of mBERT on MARC vs. conditional feature shift (upper-left; with averages marked by language), class prior shift (lower-left), and both (right). Each scatter represents a result evaluated on one subsampled target dataset.

On these datasets and language data generally, the (mis)alignment of model representations across languages could be influenced by corpus-level semantic differences besides linguistic variations. For instance, Chinese reviews in MARC are disproportionally about books. To simulate these conditions and study the effects of class prior shifts, we perform our evaluations on 500 smaller datasets subsampled from MARC with various class priors (each contains 2,500 test examples), and 700 from WikiANN.

## 2.1 INVARIANCE OF FEATURE REPRESENTATIONS

To measure representation invariance, or the alignment of class-conditioned features between two languages $p_S^{Z|Y}$, $p_T^{Z|Y}$, we compute the following quantity, referred to as *conditional feature shift*:

$$\frac{1}{k}\sum_{j=1}^{k} D(p_S^{Z|Y=j}, p_T^{Z|Y=j}), \tag{1}$$

and we use the $\ell_1$-distance between the feature means as the discrepancy measure[1],

$$D(p,q) := \left\| \mathbb{E}_{x\sim p}[x] - \mathbb{E}_{x'\sim q}[x'] \right\|_1 = \sum_{i=1}^{d} \left| \mathbb{E}_{x\sim p}[x_i] - \mathbb{E}_{x'\sim q}[x'_i] \right|. \tag{2}$$

Similar first-moment measures have appeared in prior work (Libovický et al., 2020), but the distinction here is that the effects of class prior shift are explicitly removed via conditioning, as arbitrary amounts of *unconditioned* feature shift could be generated from adjusting the class priors.

In the upper-left panel of Fig. 1, we plot the zero-shot transfer performance[2] on MARC of a fine-tuned mBERT against the conditional feature shift between English and the target language. We observe that mBERT generally transfers better to languages whose features align well with the source, and the degree of alignment between languages agrees with prior findings on cross-lingual transfer. For instance, the commonly analyzed factor of linguistic similarity is reflected by the alignment on this dataset (Karthikeyan et al., 2020): Spanish, French, and German are considered more linguistically similar to English and have better feature alignment with English than Japanese and Chinese.

The above findings suggest that achieving representation invariance is desirable for improving zero-shot cross-lingual transfer, with evidence of empirical success including the work by Cao et al. (2020) that improves mBERT performance via aligning contextualized word embeddings. But is

---

[1]We compared our first-moment measure to more sophisticated metrics such as RBF-kernel MMD (Gretton et al., 2012) and linear CKA (Kornblith et al., 2019) in preliminary studies, but found tight correlations between the results in our use case.

[2]Measured in macro-averaged $F_1$ score instead of accuracy because accuracy could be inflated on datasets with skewed priors (Azizzadenesheli et al., 2019).

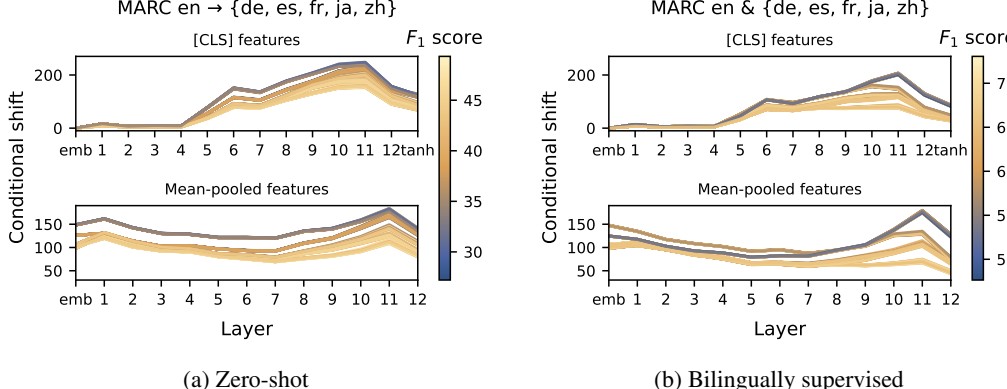

(a) Zero-shot

(b) Bilingually supervised

Figure 2: Transfer performance of mBERT on MARC under two learning settings vs. conditional shift of intermediate representations. mBERT is fine-tuned with the classification head added on top of `[CLS]` embedding, but we plot the shifts of both `[CLS]` (upper panels) and mean-pooled embeddings (lower panels). This is because `[CLS]` embeddings are not informative until they are contextualized in upper layers. Each curve represents a result evaluated on one subsampled target dataset.

invariance suitable when access to target language (unlabeled) task data is available under unsupervised transfer settings considered in prior work (Keung et al., 2019; Wu et al., 2020)? We perform the same empirical evaluation as above, but instead of testing every unsupervised approach, we fine-tune the model with supervision on both source and target languages. This is because most unsupervised objectives consist of a loss on labeled source data, and terms involving unlabeled data that are in theory linked to the true loss on target data, so their ideal objective is the supervised loss.

In Fig. 2b, we plot the conditional feature shift of mBERT trained under bilingual supervision. Unlike Fig. 1, because last-layer conditional alignment could be an artifact of having a linear classification head on top and using cross-entropy loss (Papyan et al., 2020), the shift on all intermediate layers of mBERT is shown. Compared to the zero-shot model in Fig. 2a, we see that the supervised model's better performance is accompanied by improved representation alignment across all layers. This suggests that language-invariant features are preferred over potentially language-specific ones for cross-lingual transfer, and serves as the basis for our algorithm in Section 3.

## 2.2 Distributional Shifts in Class Priors

To measure class prior shift between source and target task data, we compute the total variation between $p_S^Y$ and $p_T^Y$. In the lower-left panel of Fig. 1, we plot the zero-shot transfer performance of mBERT against class prior shifts, and observe that performance generally degrades as prior shift increases regardless of how well the features are aligned. Furthermore, we observe in Appendix A.1.1 that the degradation is aggravated when the source prior distribution is skewed, i.e. highly concentrated on a few classes.

Although prior shift is typical with real-world data, most benchmarks for cross-lingual transfer evaluation are constructed with uniform class priors so as not to shift the focus to "'tricks' for how to best handle the class imbalance" (Schwenk & Li, 2018). However, unsupervised methods that leverage unlabeled target data could be affected by prior shifts, and in cases, as we show with existing approaches in Section 4, result in worse performance than zero-shot transfer. This was not detected in prior work because evaluations were performed on class-balanced data only. Our proposed method includes an estimation and correction procedure, and our evaluation on data with prior shifts demonstrates its consistent performance even under large shifts (Section 4.2).

To conclude our analysis, we plot conditional feature shift jointly with prior shift against mBERT zero-shot transfer performance in the right panel of Fig. 1, and observe strong correlations between performance and either of the two factors when controlling the other.

## 3   Importance-Weighted Domain Alignment

For unsupervised cross-lingual learning with access to target language unlabeled task data along with labeled ones in source, the findings in the previous section suggest that good transfer performance could be achieved with class-conditional feature alignment and prior shift correction. Based on these principles, we propose class-importance-weighted domain alignment (IWDA), which closely follows the work by Tachet des Combes et al. (2020). Here, we provide a high-level description of the algorithm and highlight fixes for addressing previously undiscovered optimization instabilities. Implementation details are deferred to Appendix C.1.

Assume for now that class priors are known, and denote the true importance weights (IWs) by $w_j^* := p_T(Y = j)/p_S(Y = j)$. In Section 2.1, we demonstrated that class-conditional feature alignment, $p_S^{Z|Y} = p_T^{Z|Y}$, is desired for unsupervised transfer, but achieving this goal requires knowing the true label for every target example. Instead, we aim for a weaker goal but a necessary condition (Tachet des Combes et al., 2020), called *class-weighted* feature alignment, $p_S^{w^*, Z} = p_T^Z$, where

$$p_S^{w^*}(x, Y = j) := p_S(x|Y = j)p_S(Y = j)w_j^* = p_S(x|Y = j)p_T(Y = j).$$

If the true $w^*$ is unknown, it can be estimated when conditional feature alignment is satisfied (also known as the label shift assumption) by solving a set of equations obtained from the following first-order relation between the target prediction distribution $\hat{Y} := h(g(X))$, source confusion matrix, and $w^*$ (Saerens et al., 2002; Lipton et al., 2018):

$$p_T(\hat{Y} = i) = \sum_{j=1}^k p_T(\hat{Y} = i|Y = j)p_T(Y = j) = \sum_{j=1}^k p_S(\hat{Y} = i|Y = j)p_T(Y = j) = \sum_{j=1}^k p_S(\hat{Y} = i, Y = j)w_j^*.$$

(3)

However, if none of the assumptions are satisfied, then there is no guarantee for achieving either of the goals above without other assumptions. The hope here is that because zero-shot models are shown to be largely aligned, by alternating the optimization towards both goals, small incremental improvements to feature alignment could make prior shift estimates more accurate, and vice versa.

**Optimization.** We define the following two objectives for $g$, $h$ and $w$,

$$\min_{g,h}\left(\mathbb{E}_{(x,y)\sim p_S}[\ell(h(g(x)), y)] + \lambda \cdot D(p_S^{w,Z}, p_T^Z)\right), \text{ and } \min_w \sum_{\hat{y}=1}^k \left(p_T(\hat{y}) - \sum_{y=1}^k p_S(\hat{y}, y)w_y\right)^2.$$

The alignment objective minimizes source loss and the distributional discrepancy between $w$-weighted source features and target features. The IW estimation objective solves eq. 3 with MSE loss using current statistics from $g$, $h$, with the constraints $w \geq 0$ and $w^\top p_S^Y = 1$.

We reformulate the alignment objective into a minimax problem optimized with adversarial training, replacing $D$ with a shallow ReLU network Wasserstein-1 critic and zero-centered gradient penalty (Arjovsky et al., 2017; Thanh-Tung et al., 2019). We found that Wasserstein-1 adversarial loss responds to importance weighting better than cross-entropy loss (Ganin et al., 2016).

**Improving Stability of IW Estimates.** In our preliminary experiments with the IW estimation procedure by Tachet des Combes et al. (2020), which solves eq. 3 using quadratic programming once every epoch, we found that the estimates always tend to 1 in long training episodes (i.e. no shift is estimated) although prior shifts are present, and even if they were accurate in previous iterations. This issue was not previously documented, and we speculate the cause to be the noise and fluctuations from the optimization procedure for feature alignment. Another behavior we found is the tendency of overestimating the majority class of the target data.

To address these instabilities, we include three fixes in our implementation while leaving further investigations to future work. We (1) update $w$ at every training step with projected gradient descent for always keeping it close to optimality, (2) use a decaying learning rate schedule for $w$, and (3) add $\ell_2$-regularization to $w$ (Azizzadenesheli et al., 2019).

## 4 EXPERIMENTS

We apply importance-weighted domain alignment (IWDA) to fine-tune mBERT (cased) and XLM-R Large for unsupervised cross-lingual learning, and compare it to baseline methods based on two semi-supervised learning (SSL) techniques. The NLP tasks are sentiment analysis, named-entity recognition (NER), and textual entailment. As with most work on cross-lingual transfer, English is used as the source language in our experiments.

In addition to standard multilingual benchmarks where the priors are (largely) the same across languages (Section 4.1), we also use datasets subsampled with varying priors as in Section 2 to evaluate performance under prior shifts and investigate the limitations of existing approaches (Section 4.2).

**Datasets.** For **NER**, two multilingual benchmarks are used: CoNLL-2002 & 2003 (Tjong Kim Sang, 2002; Tjong Kim Sang & De Meulder, 2003) and WikiANN (Pan et al., 2017; Rahimi et al., 2019). The former includes English, German, Spanish and Dutch, and the latter covers over 200 languages, from which a subset of 40 in the XTREME benchmark is selected (Hu et al., 2020). Although prior shifts (relative to English) are present in these datasets, they are mostly mild except for some languages in WikiANN. For **sentiment analysis**, the Multilingual Amazon Reviews Corpus (MARC) is used (Keung et al., 2020), which consists of product ratings and reviews in English, German, Spanish, French, Japanese, and Chinese. For **textual entailment**, the XNLI dataset is used (Conneau et al., 2018), where each example is a sentence pair and the task is to determine whether the first sentence entails or contradicts the second, or neither. The test and validation examples are human-translated from English into 14 languages, and the training examples are machine-translated (Conneau & Lample, 2019).

**Baselines.** **Knowledge distillation (KD)** transfers knowledge from the source to the target domain by first training a teacher model on labeled source data then training a student model to mimic the teacher outputs on unlabeled target data (Hinton et al., 2015). A variant of KD was evaluated on NER by Wu et al. (2020), whose modifications include freezing the bottom three layers of the teacher and student mBERTs (Wu & Dredze, 2019). **Self-training (ST)** trains the model on source labeled data first, then iteratively assigns pseudo-labels to unlabeled target data and trains on the most confident predictions (Nigam & Ghani, 2000). Our implementation is based on Dong & de Melo (2019), which was evaluated on sequence classification tasks with mBERT. While these baselines are not exhaustive, they are representative in that most methods for unsupervised cross-lingual transfer involve procedures that share the same fundamental ideas. For instance, Bari et al. (2021) combined data augmentation with an extension of self-training, called tri-training (Zhou & Li, 2005).

For **IWDA**, a two-stage procedure is applied, where we first fine-tune the model on labeled source data only (zero-shot learning), then continue with the IWDA objectives on the source and unlabeled target task data, both for four epochs. We also evaluate the effectiveness of class-weighted feature alignment by assuming knowledge of target class priors, and label the results **IWDA (oracle)**.

The hyperparameter settings are included in Appendix C.2. Due to space constraints, we present results from mBERT on sentiment analysis and NER in this section while deferring results on XNLI and from XLM-R in Appendix A.2 and B. Unless otherwise noted, results are from our implementation.

### 4.1 RESULTS ON STANDARD BENCHMARKS

Results with mBERT for sentiment analysis on MARC are presented in Table 1, and those for NER on CoNLL datasets and three low-resource languages on WikiANN are presented in Tables 2 and 3, respectively. The remaining WikiANN results are included in Table 4 in the appendix. All IWDA results show improvements upon the zero-shot baselines, and are accompanied by decreases in conditional feature shift. The average decreases (measured with eqs. 1 and 2, with 100% being perfect alignment) are 51.53% and 64.77% (oracle) on MARC, 21.60% and 22.33% (oracle) on CoNLL, and 32.91% and 36.34% (oracle) on WikiANN.

IWDA, however, sometimes fall short of SSL baselines, which is likely due to the noise from the optimization and adversarial training as well as some amounts of incorrect alignment that cannot generally be prevented, as discussed in Section 3. One way to reduce noise is to improve optimality

Table 1: mBERT on MARC (in accuracy).

|  | en | de | es | fr | ja | zh |
|---|---|---|---|---|---|---|
| *Supervised and zero-shot baselines* | | | | | | |
| Supervised (in-language) | 58.50 | 61.19 | 57.76 | 57.05 | 57.97 | 53.74 |
| Zero-shot | 58.50 | 44.80 | 46.49 | 46.02 | 37.37 | 38.48 |
| *Unsupervised results* | | | | | | |
| Wu et al., 2020 | - | 47.43 | 49.56 | 49.06 | 35.77 | 38.54 |
| ST | 58.25 | 50.74 | 48.80 | 47.96 | 42.10 | 41.40 |
| IWDA | 56.87 | 51.94 | 49.77 | 49.78 | 42.62 | 44.04 |
| + KD | 58.55 | **54.02** | **51.74** | **51.67** | **45.14** | 45.25 |
| + ST | 56.41 | 53.11 | 51.00 | 49.91 | 43.59 | **45.27** |
| IWDA (oracle) | 55.96 | 51.95 | 50.83 | 50.01 | 44.91 | 45.96 |

Table 2: mBERT on CoNLL (in $F_1$).

|  | en | de | es | nl |
|---|---|---|---|---|
| *Supervised and zero-shot baselines* | | | | |
| Supervised (in-language) | 91.97 | 82.82 | 87.38 | 90.94 |
| Zero-shot | 90.57 | 69.77 | 74.14 | 78.28 |
| *Unsupervised results* | | | | |
| Keung et al., 2019 | - | 71.9 | 74.3 | 77.6 |
| Wu et al., 2020 | - | 73.22 | 76.94 | **80.89** |
| IWDA | 90.77 | 72.56 | 76.11 | 78.63 |
| + KD | 90.89 | **73.71** | **77.14** | 79.72 |
| IWDA (oracle) | 90.75 | 72.58 | 76.48 | 79.17 |

Table 3: mBERT on WikiANN (in $F_1$).

|  | en | bn | my | ur |
|---|---|---|---|---|
| *Supervised and zero-shot baselines* | | | | |
| Supervised (in-language) | 83.73 | 92.69 | 69.08 | 93.29 |
| Zero-shot | 83.73 | 68.55 | 50.16 | 35.78 |
| *Unsupervised results* | | | | |
| Wu et al., 2020 | - | 71.78 | 46.32 | 44.52 |
| IWDA | 83.93 | 71.05 | 55.86 | 75.79 |
| + KD | 83.07 | **73.15** | **56.05** | **77.58** |
| IWDA (oracle) | 83.93 | 72.19 | 60.22 | 74.98 |

of the adversarial critic throughout training with more updates and passes over the data, and with which we observe improved performance on CoNLL (Table 5). Another way to reduce noise is to follow IWDA with a final stage SSL fine-tuning because of their label smoothing effects (Yuan et al., 2020). Alternatively, we can view IWDA as a pre-processing step for SSL, since they theoretically rely on the assumption that source and target data are identically distributed, which IWDA helps satisfy with class-weighted feature alignment. These experiments are labeled by **IWDA + KD** and **IWDA + ST**, and mostly achieve the best results among unsupervised methods (in bold).

Finally, we note that the performance from applying IWDA alone depends on (1) whether pre–trained representations are already partially aligned, which is shown for mBERT in Section 2, and (2) whether the benefits of enhancing the feature alignment outweigh the noise from IWDA and the potential small amounts of misalignment. The latter may not always hold, especially on larger models with better zero-shot transferability. For instance, zero-shot learning with XLM-R Large on MARC achieves better performance and feature alignment than mBERT, and although applying IWDA reduces conditional feature shift, results are worse than zero-shot baselines (Table 7). In these cases, we expect the utility of IWDA to be realized when combined with additional resources, but leave investigations to future work. For instance, Bari et al. (2021) achieved good performance on XLM-R with data augmentation that leverages the multilingual mask-filling ability of XLM-R.

## 4.2 RESULTS UNDER CLASS PRIOR SHIFTS

In this section, we evaluate IWDA and SSL methods on mBERT under class prior shift between source and target task data by performing unsupervised transfer from English to Japanese on MARC and from English to German on CoNLL, using the same datasets subsampled with varying priors in Section 2. The results relative to zero-shot baselines are plotted in Fig. 3.

For IWDA, we observe consistent improvements across the range of prior shifts. SSL approaches are also performant under mild to no shifts, but quickly deteriorate as the shift increases, eventually resulting in negative transfer. This failure mode is not due to prior shift alone, but also the lack of conditional alignment between source and target data that SSL assumes. As demonstrated with IWDA + ST and KD results, when feature alignment is improved, subsequent SSL training can deliver better performance and avoid negative transfer under large shifts.

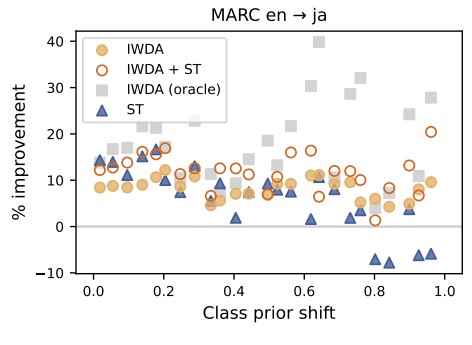 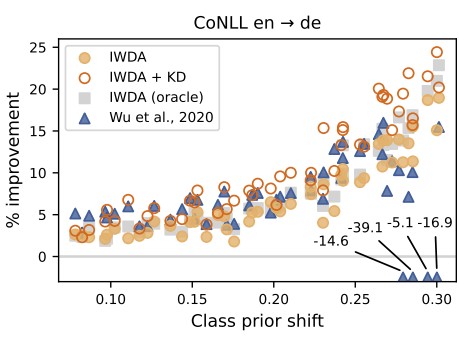

(a) MARC English to Japanese            (b) CoNLL English to German

Figure 3: Percent improvement in mBERT transfer performance (macro-averaged $F_1$) over zero-shot baselines (0%) when applying unsupervised methods under presence of class prior shifts, evaluated on datasets subsampled from MARC and CoNLL with varying priors.

We note that the importance weighting (IW) procedure in IWDA is integral to its performance under the presence of any amount of prior shift, and any alignment-based methods that do not account for it will result in negative transfer. This is because under prior shift, unweighted feature alignment means that some target features from one class must be aligned with source features from a different class, hence any classifier trained on the source will make mistakes on those misaligned target features (Zhao et al., 2019). We demonstrate this on CoNLL English to German transfer in Fig. 7 in the appendix, where IWDA without IW sees decreasing performance as training progresses even though the shift is mild.

Interestingly, Keung et al. (2019) apply an alignment-based method to CoNLL NER without IW that does not cause negative transfer (Table 2). This is because their method does not align the token embeddings that are used as features by the linear classification head, but the `[CLS]` embeddings. Since the `[CLS]` embeddings contain context information computed from the (intermediate) token embeddings, their alignment can be viewed as a weak alignment of the features. Although it avoids the negative transfer from prior shifts, it also misses opportunities for greater performance gains, as comparisons with IWDA results show.

Finally, we report the accuracy of the IW estimates. The average percent prior shift corrections, computed as the decrease in total variation (TV) between true target priors and final estimated priors normalized by the amount of shift (with 100% being perfectly accurate), are 87.33% on CoNLL and 28.08% on MARC for shifts over 0.8 in TV. For shifts under 0.8, the estimates are often worse than simply (although incorrectly) assuming a uniform prior, with a -335.8% in correction. This is because MARC reviews have noisy labels (as one could waver between giving a 3- or 4-star rating), and datasets with more class balance are harder to train on even with supervision, so the performance of the IW estimator could be affected by bad source confusion matrices (Lipton et al., 2018).

## 5 ADDITIONAL RELATED WORK

**Cross-Lingual Ability of Multilingual LMs.** Most work investigating the cross-lingual ability of mBERT perform analysis via probing experiments, through which they identify factors that affect transfer performance that include linguistic similarity, size of pre-training corpus, domain similarity, parameter sharing, and model depth (Pires et al., 2019; Karthikeyan et al., 2020; Lauscher et al., 2020; Conneau et al., 2020b; Dufter & Schütze, 2020). Another line of research, including the present work, examines the intermediate representations and observes that mBERT learns cross-lingually shared representations that contribute to transfer performance (Wu & Dredze, 2019; Conneau et al., 2020b; Muller et al., 2021).

**Language-Invariant Representations.** The idea of learning language-invariant representations is not new and has served as the basis for a long history of approaches for cross-lingual transfer on neural LMs.

For static word embedding models, language-invariance is achieved by aligning the embeddings of bilingual dictionary pairs (Mikolov et al., 2013; Smith et al., 2017). With adversarial training, the dictionary need not be provided as it could be automatically induced (Zhang et al., 2017; Artetxe & Schwenk, 2019; Patra et al., 2019; Dubossarsky et al., 2020), and this approach has powered many unsupervised machine translation approaches (Artetxe et al., 2018; Lample et al., 2018; Arivazhagan et al., 2019; Pham et al., 2019; Hu et al., 2021). For deep models such as mBERT, language-invariance is achieved with a cross-lingual feature representation induced by sharing the transformer body (Artetxe et al., 2020b; Pfeiffer et al., 2020).

Under unsupervised and supervised settings, prior work has proposed cross-lingual learning techniques based on representation alignment for models ranging from static word embeddings (Joty et al., 2017; Chen et al., 2018), RNN (Kim et al., 2017; Huang et al., 2019), to mBERT (Keung et al., 2019; Zheng & Lapata, 2019; Xia et al., 2021). For a survey, see Ruder (2019).

**Domain Adaptation.** Our method, based on the work of Tachet des Combes et al. (2020), is initially motivated by the problem of (unsupervised) domain adaptation, a framework for handling domain shifts between training and test data (Ben-David et al., 2007). For example, training a POS tagger on news articles with the goal of deploying it on biomedical papers, whose vocabulary is significantly different.

For unsupervised cross-lingual learning, we may apply adaptation techniques proposed for domain shift for cross-lingual transfer by treating task data in each language as a distinct domain. Besides alignment-based methods (Li et al., 2020; Vernikos et al., 2020), including ours, another family of approaches is semi-supervised learning, such as self-training (Dong & de Melo, 2019), tri-training (Ruder & Plank, 2018), knowledge distillation (Wu et al., 2020), and data augmentation (Wang et al., 2018; Maharana & Bansal, 2020).

## 6 Conclusion and Future Work

Motivated by the question of *how shared representations contribute to cross-lingual learning* on multilingual neural LMs, we perform empirical analyses that showed that downstream cross-lingual transfer performance is strongly correlated with the invariance of feature representations and negatively affected by the class prior shift between source and target task data. Based on these findings, we propose and evaluate importance-weighted domain alignment (IWDA) for unsupervised cross-lingual transfer, and show its effectiveness for unsupervised transfer and superiority over semi-supervised learning (SSL) methods under large prior shifts. Furthermore, by combining IWDA and SSL, further performance gains are achieved.

While our present implementation of IWDA is largely effective, it could be improved with future work on unsupervised domain adaptation, and techniques for prior shift estimation and distribution alignment, including or beyond adversarial training. Following the results from combining IWDA and SSL, further experiments could include combining it with more elaborate approaches for unsupervised cross-lingual transfer. Finally, the success of existing zero-shot and unsupervised methods is contingent upon the cross-lingual representations acquired by the pre-trained LMs without supervision. Fundamental improvements to their transfer require continued investigation on the emergence of their cross-lingual ability.

### Acknowledgments

We thank Keyulu Xu and Mozhi Zhang for helpful discussions, and the anonymous reviewers for valuable comments. HZ would like to thank the support from a Facebook research award.

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

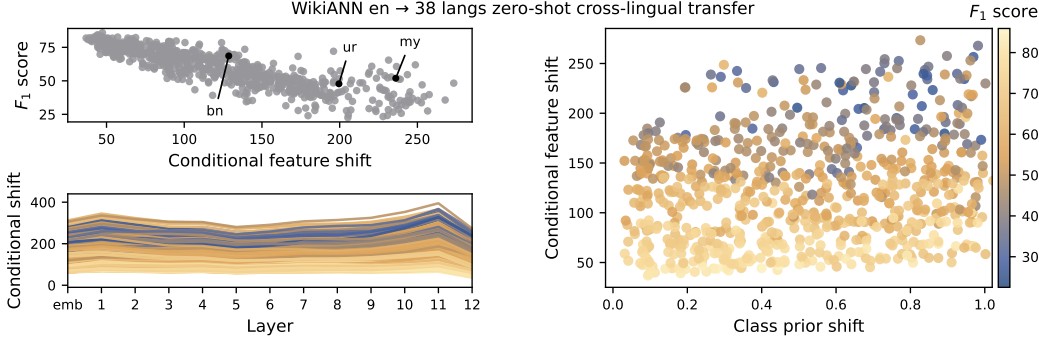

Figure 4: Zero-shot transfer performance of mBERT on WikiANN vs. conditional shift of intermediate representations (lower-left), final-layer features (upper-left; with averages marked by language), and jointly with class prior shift (right).

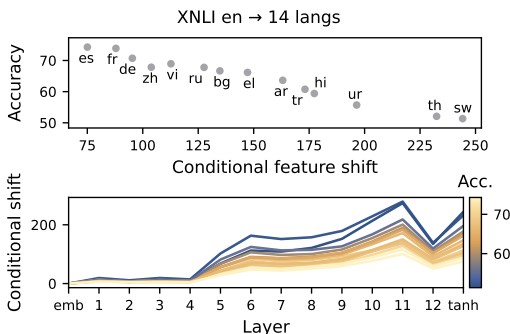

Figure 5: Zero-shot transfer performance of mBERT on XNLI vs. conditional shift of intermediate representations (lower-left), and final-layer features (upper-left; with averages marked by language). The [CLS] embeddings are examined.

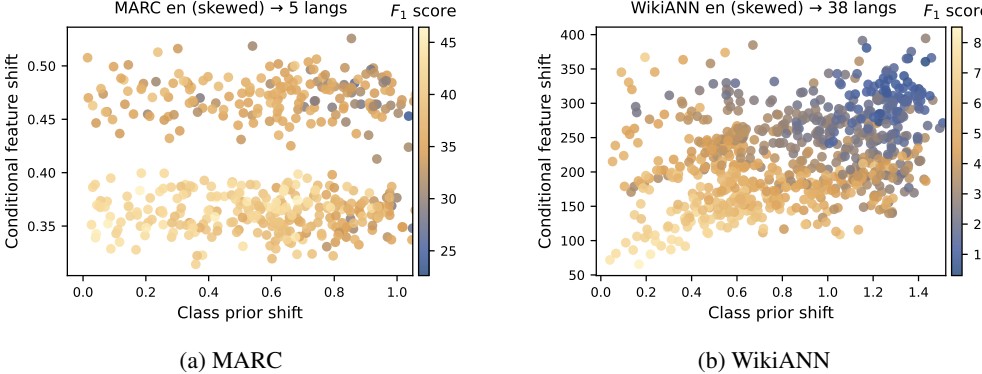

(a) MARC                    (b) WikiANN

Figure 6: Zero-shot transfer performance of mBERT on two datasets vs. conditional feature shift jointly with class prior shift. **Models are trained on source data with skewed prior distributions.**

## A    ADDITIONAL MBERT RESULTS

### A.1    ADDITIONAL MBERT ANALYSES

In Section 2, we demonstrate that transfer performance of mBERT is strongly correlated with feature alignment and (negatively) class prior shift between source and target task data. Here, we show that these observations also hold when transferring to NLP tasks besides sentiment analysis.

**Named-Entity Recognition (NER).** We use the WikiANN dataset, and analysis is performed on 700 smaller datasets subsampled from the original WikiANN with varying priors. As the 39 languages in this datasets have distinct linguistic properties and different resourcefulness (a summary of the languages can be found in Table 5 of Hu et al. (2020), and the sizes their pre-training data in Table 1 of Wu & Dredze (2020)), results based on it shall be less influenced by confounding effects, if any.

The results are provided in Fig. 4. We observe the same correlations, but they are weaker between the prior shift and the performance. This is likely because NER is a structured prediction problem where mBERT could leverage dependencies between words for its predictions, making it robust to prior shifts. However, the effects of prior shift will become pronounced if the source data have a skewed prior distribution, which we show below in Fig. 6b and discuss in Section A.1.1.

**Textual Entailment.** For this high-level semantic task, we use the XNLI dataset, where we train the model on the English portion and evaluate on human-translated validation data in 14 target languages. We do not subsample XNLI because of the small data size, hence prior shift results are not included. The results are presented in Fig. 5.

### A.1.1    SOURCE DATA WITH SKEWED PRIORS

So far in our analyses, we have only evaluated models that trained on relatively class-balanced source datasets. We show that if the class prior distribution of the source data is skewed, then the negative effects of prior shift on transfer performance are more prominent.

For this study, we use MARC. We generate a skewed English dataset via subsampling according to the prior distribution of 14.86%, 6.43%, 8.03%, 19.28%, and 51.41%[3], corresponding to the proportion of 1- to 5-star reviews. We then train mBERT on this dataset and evaluate its performance on subsampled target datasets. The results are presented in Fig. 6a, where the correlation between performance and prior shift is much stronger than when the source is class-balanced (Fig. 1).

We also evaluate on WikiANN. Although its original English portion is already considerably skewed (with class ○ making up 50.75% of the labels), its influence is observed to be mild likely because

---

[3]This is the actual distribution of product ratings on Amazon.com (McAuley & Leskovec, 2013).

Table 4: mBERT on WikiANN (in $F_1$).

|  | en | af | ar | bg | bn | de | el | es | et | eu |
|---|---|---|---|---|---|---|---|---|---|---|
| Supervised | 83.73 | 91.36 | 87.62 | 92.64 | 92.69 | 89.47 | 90.55 | 91.42 | 91.58 | 91.72 |
| Zero-shot | 83.73 | **77.63** | **40.68** | 77.89 | 68.55 | **79.01** | 71.19 | 76.73 | 76.74 | 59.76 |
| IWDA | 83.93 | 77.50 | 37.59 | **78.57** | **71.05** | 78.86 | **72.19** | **83.37** | 79.16 | 67.11 |
| IWDA (oracle) | 83.93 | 77.18 | 37.29 | 79.65 | 72.19 | 79.07 | 72.13 | 83.72 | 80.23 | 72.69 |

|  | fa | fi | fr | he | hi | hu | id | it | ja | jv |
|---|---|---|---|---|---|---|---|---|---|---|
| Supervised | 92.11 | 91.20 | 90.15 | 85.19 | 86.10 | 92.67 | 91.88 | 91.44 | 73.18 | 73.71 |
| Zero-shot | **40.27** | 77.87 | **80.47** | 56.55 | **66.08** | **76.55** | 60.67 | 80.67 | 27.49 | **61.51** |
| IWDA | 28.32 | **78.43** | 79.99 | **56.57** | 64.37 | 76.00 | 51.35 | 80.46 | **38.05** | 61.48 |
| IWDA (oracle) | 46.33 | 79.56 | 83.78 | 57.62 | 67.01 | 76.69 | 72.00 | 80.99 | 34.98 | 60.57 |

|  | ka | kk | ko | ml | mr | ms | my | nl | pt | ru |
|---|---|---|---|---|---|---|---|---|---|---|
| Supervised | 86.55 | 85.06 | 87.71 | 82.68 | 85.42 | 93.27 | 69.08 | 91.37 | 91.45 | 88.28 |
| Zero-shot | 66.41 | 46.87 | 60.52 | 53.33 | 57.97 | 67.63 | 50.16 | **82.16** | 80.07 | 65.61 |
| IWDA | **67.14** | **48.15** | **61.18** | **53.75** | **61.80** | **69.78** | **55.86** | 81.03 | **82.40** | **66.00** |
| IWDA (oracle) | 67.39 | 49.11 | 63.29 | 59.47 | 61.55 | 69.01 | 60.22 | 81.05 | 83.66 | 72.00 |

|  | sw | ta | te | th[4] | tl | tr | ur | vi | yo | zh |
|---|---|---|---|---|---|---|---|---|---|---|
| Supervised | 88.98 | 83.53 | 78.53 | 76.84 | 93.64 | 92.35 | 93.29 | 90.83 | 82.93 | 81.14 |
| Zero-shot | **69.35** | 50.22 | 50.63 | 1.31 | 93.57 | 73.78 | 35.78 | 71.19 | 49.29 | 43.29 |
| IWDA | 67.56 | **53.93** | **52.09** | **1.60** | 75.16 | 74.36 | 75.79 | 80.20 | 52.07 | **49.90** |
| IWDA (oracle) | 67.40 | 55.46 | 53.84 | 8.39 | 76.50 | 78.64 | 74.98 | 78.85 | 54.60 | 56.26 |

Table 5: mBERT on CoNLL NER (in $F_1$) when the adversary enjoys better optimization. Results in bold are best among unsupervised methods including those in Table 2.

|  | en | de | es | nl |
|---|---|---|---|---|
| IWDA | 91.19 | 72.91 | 76.93 | 78.62 |
| + KD | 90.68 | **73.83** | **77.87** | 79.96 |
| IWDA (oracle) | 91.32 | 72.47 | 77.24 | 79.31 |

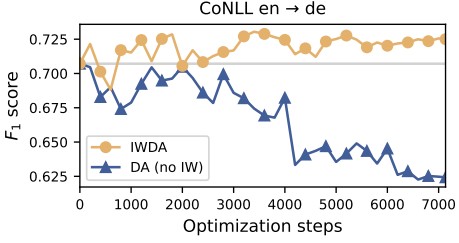

Figure 7: mBERT transfer performance on CoNLL from English to German vs. update steps. Comparing IWDA to without IW.

NER is a structured prediction problem as discussed above (Fig. 4). We skew its prior distribution by keeping only sequences containing at least one O label, increasing its proportion to 80.92%. The results of mBERT trained on this dataset are presented in Fig. 6b, where we now see pronounced effects from prior shifts on transfer performance.

## A.2 ADDITIONAL mBERT EXPERIMENTS

In Table 4, we present IWDA results on WikiANN in the 36 target languages not included in Table 3.

In Table 5, we present improved IWDA results on CoNLL when the adversary enjoys better optimization with more updates (we alternate between four adversary steps and one representation alignment step instead of performing simultaneous updates) and passes over the data (training took 40 passes over the data). We decrease `lambda_iw` from 2 to 1.

In Fig. 7, we show that IWDA without IW would cause negative transfer under prior shifts as discussed in Section 4.2, even under the mild shift between English and German on CoNLL.

**Textual Entailment.** In addition to the NLP tasks of sentiment analysis and NER on which IWDA is evaluated in Section 4, we also perform evaluations on the high-level semantic task of textual entailment using the XNLI dataset.

---

[4]The bad zero-shot and unsupervised performance on Thai is due differences in the tokenization schemes used by mBERT and WikiANN dataset creators, who split diacritical characters into multiple tokens.

Table 6: mBERT on non-benchmark XNLI (in accuracy).

|  | en | ar | bg | de | el | es | fr | hi | ru | sw | th | tr | ur | vi | zh |
|---|---|---|---|---|---|---|---|---|---|---|---|---|---|---|---|
| *Supervised and zero-shot baselines* | | | | | | | | | | | | | | | |
| Supervised (bilingual) | 73.60 | 65.32 | 68.58 | 68.94 | 66.32 | 71.18 | 69.28 | 63.02 | 67.66 | 57.32 | 63.36 | 63.02 | 48.60 | 68.24 | 69.62 |
| Zero-shot | 73.60 | 60.64 | 60.60 | 67.12 | **63.14** | 67.58 | 67.06 | 57.30 | 64.30 | 47.28 | 51.40 | 57.36 | 43.16 | 62.98 | 63.70 |
| *Unsupervised learning* | | | | | | | | | | | | | | | |
| ST | 73.04 | 60.20 | 63.40 | 66.30 | 62.00 | 67.92 | 67.16 | 56.62 | 64.32 | 47.76 | 53.40 | 57.80 | 46.04 | 62.88 | 63.90 |
| IWDA | 73.03 | **61.16** | 64.16 | **67.30** | 62.68 | **68.28** | **67.34** | **57.92** | **65.06** | 47.78 | **54.80** | **58.00** | 44.24 | **63.60** | 65.08 |
| + ST | 73.10 | 60.28 | **64.18** | 66.66 | 62.44 | 68.02 | 66.96 | 57.70 | 64.56 | **48.26** | 53.78 | 57.92 | **46.94** | 63.40 | **65.12** |
| IWDA (oracle) | 72.94 | 60.74 | 64.62 | 67.02 | 63.22 | 68.88 | 67.06 | 58.88 | 64.84 | 45.96 | 54.50 | 59.36 | 46.76 | 64.28 | 65.82 |

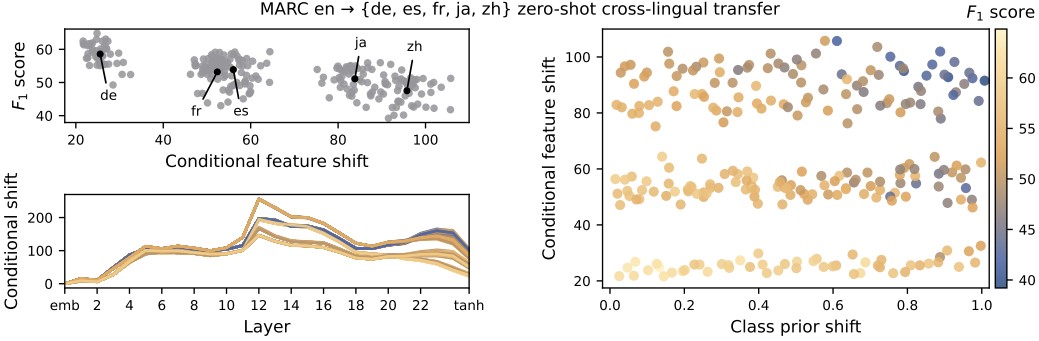

Figure 8: Zero-shot transfer performance of XLM-R on MARC vs. conditional shift of intermediate representations (lower-left), final-layer features (upper-left; with averages marked by language), and jointly with class prior shift (right). Each curve and scatter represents a result on one subsampled dataset, and the `[CLS]` embeddings are examined.

Non-English XNLI test data are human-translated from English, but the training data are only available in machine translations (Conneau et al., 2020a). Since IWDA training requires unlabeled target task data, to avoid confounding effects due to data parallelism and test-train domain mismatch from machine-translation artifacts (Artetxe et al., 2020a), we partition machine-translated data into equal portions of size 20,000 (5% of the original size) and randomly select disjoint partitions to use as source and target data, as done in (Bari et al., 2021). The results, which are not benchmark due to the setup, are included in Table 6.

## B    XLM-R RESULTS

Besides mBERT, we also include some analysis and experiment results with XLM-R Large, a much larger model pre-trained on more data than mBERT and achieving state-of-the-art cross-lingual learning performance.

In Fig. 8 on the next page, by evaluating on MARC, we show that the correlations between feature alignment, prior shift and performance also hold on XLM-R.

In Table 7, we present IWDA and SSL results when applied to XLM-R on MARC. A discussion on these results is included in Section 4.1. In Tables 8, 9, and 10, we present results on CoNLL, WikiANN (low-resource languages), and XNLI, respectively.

In Fig. 9, we present IWDA and SSL results on MARC English to Japanese transfer and CoNLL English to German under prior shifts. We see that IWDA also outperforms SSL with XLM-R under large prior shifts, where the latter often causes negative transfer.

Table 7: XLM-R on MARC (in accuracy).

|  | en | de | es | fr | ja | zh |
|---|---|---|---|---|---|---|
| *Supervised and zero-shot baselines* | | | | | | |
| Supervised (in-language) | 62.05 | 64.66 | 60.70 | 59.40 | 60.40 | 56.24 |
| Zero-shot | 62.05 | 62.52 | **59.04** | 56.43 | 54.32 | 51.42 |
| *Unsupervised results* | | | | | | |
| Wu et al., 2020 | - | **62.96** | 57.50 | 56.50 | 53.38 | 50.90 |
| ST | 61.00 | 62.32 | 58.09 | 56.62 | 55.27 | 52.10 |
| IWDA | 60.21 | 61.13 | 56.93 | 56.27 | 54.87 | 50.57 |
| + KD | 60.26 | 62.42 | 58.60 | **57.24** | **56.94** | **52.20** |
| + ST | 60.05 | 61.59 | 57.52 | 56.05 | 55.43 | 51.73 |
| IWDA (oracle) | 59.99 | 61.33 | 57.65 | 56.20 | 56.27 | 52.39 |

Table 8: XLM-R on CoNLL (in $F_1$).

|  | en | de | es | nl |
|---|---|---|---|---|
| *Supervised and zero-shot baselines* | | | | |
| Supervised (in-language) | 92.92 | 85.81 | 89.72 | 92.53 |
| Zero-shot | 92.49 | 72.37 | 77.06 | 82.03 |
| *Unsupervised results* | | | | |
| Wu et al., 2020 | - | 73.68 | 78.52 | 81.86 |
| Bari et al., 2021 | - | **80.99** | **83.24** | **85.32** |
| IWDA | 92.55 | 75.42 | 79.08 | 81.79 |
| + KD | 92.26 | 76.66 | 79.43 | 82.05 |
| IWDA (oracle) | 92.42 | 75.32 | 78.94 | 81.82 |

Table 9: XLM-R on WikiANN (in $F_1$).

|  | en | bn | my | ur |
|---|---|---|---|---|
| *Supervised and zero-shot baselines* | | | | |
| Supervised (Hu et al., 2020) | 84.7 | 97.8 | 76.8 | 97.1 |
| Zero-shot | 83.55 | 76.24 | 58.12 | 62.66 |
| *Unsupervised results* | | | | |
| Wu et al., 2020 | - | 73.49 | 51.60 | 56.05 |
| Bari et al., 2021 | - | **82.68** | **70.61** | **84.99** |
| IWDA | 83.44 | 78.59 | 55.37 | 76.27 |
| + KD | 81.34 | 79.40 | 55.87 | 78.03 |
| IWDA (oracle) | 83.49 | 77.22 | 60.00 | 80.17 |

Table 10: XLM-R on non-benchmark XNLI (in accuracy).

|  | en | ar | bg | de | el | es | fr | hi | ru | sw | th | tr | ur | vi | zh |
|---|---|---|---|---|---|---|---|---|---|---|---|---|---|---|---|
| *Supervised and zero-shot baselines* | | | | | | | | | | | | | | | |
| Supervised (bilingual) | 84.36 | 77.48 | 79.08 | 80.42 | 80.26 | 81.18 | 80.16 | 70.82 | 78.24 | 68.14 | 77.24 | 74.02 | 49.68 | 77.20 | 77.08 |
| Zero-shot | 84.36 | 74.72 | 77.42 | 78.58 | **77.68** | 80.26 | 79.14 | **68.12** | 76.84 | 64.30 | 73.82 | **72.30** | 45.82 | **74.92** | 75.20 |
| *Unsupervised learning* | | | | | | | | | | | | | | | |
| ST | 84.35 | 74.78 | **78.14** | 78.10 | 77.60 | 79.48 | 79.26 | 67.20 | 76.90 | 64.60 | 73.94 | 72.00 | **47.26** | 73.82 | 74.54 |
| IWDA | 84.61 | 74.31 | 77.32 | 78.36 | 77.22 | 80.14 | **79.54** | 67.92 | **76.94** | 64.20 | 74.12 | 71.64 | 45.70 | 74.44 | 74.94 |
| + ST | 84.51 | **75.62** | 77.36 | **78.82** | 77.46 | 79.94 | 79.20 | 67.94 | 76.84 | **65.70** | 74.14 | **72.30** | 47.12 | 74.14 | **75.86** |
| IWDA (oracle) | 84.69 | 74.68 | 77.98 | 78.58 | 78.22 | 79.78 | 79.74 | 68.24 | 76.90 | 64.56 | 74.14 | 72.84 | 46.98 | 75.20 | 76.10 |

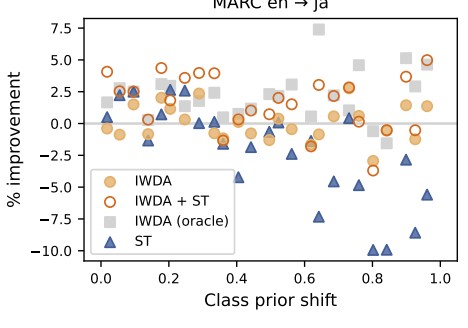

(a) MARC English to Japanese

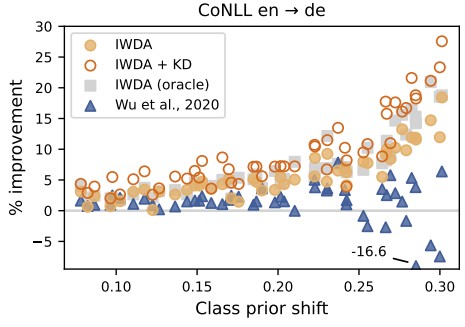

(b) CoNLL English to German

Figure 9: Percent improvement in XLM-R transfer performance (macro-averaged $F_1$) over zero-shot baselines (0%) when applying unsupervised methods under presence of class prior shifts, evaluated on datasets subsampled from MARC and CoNLL with varying priors.

## C   IMPLEMENTATION DETAILS

### C.1   ADDITIONAL IWDA DETAILS

This section complements the high-level description of IWDA given in Section 3 with more details on the implementation.

Recall the alignment and importance weight (IW) estimation objectives from Section 3, reproduced below:

$$
\min_{g,h} \left( \mathbb{E}_{(x,y) \sim p_S}[\ell(h(g(x)), y)] + \lambda \cdot D(p_S^{w,Z}, p_T^Z) \right), \text{ and } \min_w \sum_{\hat{y}=1}^k \left( p_T(\hat{y}) - \sum_{y=1}^k p_S(\hat{y}, y) w_y \right)^2.
$$
(4)

**Alignment Objective.**   The alignment objective contains a discrepancy term between importance-weighted source and target feature distributions. The source task loss is added to prevent $g$ from degenerating to a trivial representation, e.g. constant maps $x \mapsto c$.

We reformulate it as a minimax problem by replacing the explicit evaluation of the discrepancy $D$ with a critic function $f$ (also called the adversary), as in generative adversarial networks (GAN) literature (Ganin et al. 2016; Goodfellow et al. 2014). The role of the adversary is to distinguish features $z := g(x)$ computed on examples $x$ sampled from the source domain from those on target examples, by outputting scores for predicting the label of the originating domain (w.l.o.g. source inputs are labeled 0). The adversary would not distinguish source and target features when they are well-aligned.

The above turns the alignment objective into

$$
\min_{g,h} \left( \mathbb{E}_{(x,y) \sim p_S}[\ell(h(g(x)), y)] + \lambda \max_{f \in \mathcal{F}} \left( \underbrace{\mathbb{E}_{x \sim p_S^{w,X}}[\ell_{\mathrm{ad}}(f(g(x)), 0)] - \mathbb{E}_{x \sim p_T^X}[\ell_{\mathrm{ad}}(f(g(x)), 1)]}_{(\triangle)} \right) \right),
$$

where $\ell_{\mathrm{ad}}$ is a specific choice of adversarial loss accompanied by an adversary function class $\mathcal{F} \ni f$. The inputs to this loss function are the scores computed by the adversary and the domain labels of the examples.

In our implementation, we use Wasserstein-1 loss for $\ell_{\mathrm{ad}}$, whose accompanying adversary function class $\mathcal{F}$ is the set of 1-Lipschitz functions (Arjovsky et al., 2017). It is shown that when $f$ attains its maximum, the term $(\triangle)$ exactly computes the Wasserstein-1 distance between $p_S^{w,Z}$ and $p_T^Z$. We parameterize $\mathcal{F}$ as a ReLU network with one hidden layer of width 2048. The Lipschitz condition is enforced using soft constraints in the form of zero-centered gradient penalty, which penalizes the gradient norm of $f$ evaluated at randomly sampled points $\tilde{z} \in \mathcal{Z}$ interpolating between pairs of source and target features (Thanh-Tung et al., 2019).

For added stability, we condition the alignment of features on their pseudo-labels, using the method called CDAN proposed by Long et al. (2018). This method replaces the original input to the adversary, $z := g(x) \in \mathcal{Z}$, by the outer product between $z$ and the classifier output distribution, namely $g(x) \otimes h(g(x)) \in \mathbb{R}^{dk}$.

Putting everything together—minimax reformulation, Wasserstein-1 loss, zero-centered gradient penalty, and conditional alignment—the empirical alignment loss computed on a mini-batch of source examples $(x_i, y_i)_{i=1}^n$ and target examples $(x_j')_{j=1}^m$ is

$$
\frac{1}{n} \sum_{i=1}^n \ell(h(g(x_i)), y_i) + \lambda \left( \frac{1}{n} \sum_{i=1}^n f(g(x_i) \otimes h(g(x_i))) - \frac{1}{m} \sum_{j=1}^m f(g(x_j') \otimes h(g(x_i'))) \right) - \frac{\mu}{\tilde{n}} \sum_{i=1}^{\tilde{n}} \|(\nabla f)(\tilde{z}_i)\|^2,
$$

where $\tilde{n} := \min(n, m)$ and $\tilde{z}_i$ is a random point on the line connecting $g(x_i)$ and $g(x_i')$. Here, two hyperparameters are introduced: $\lambda$ (`lambda_da`) controls the strength of alignment, and $\mu$ (`lambda_gp`) controls the strength of enforcing the Lipschitz condition.

**IW Estimation Objective.** Our proposed modifications for improving its stability in Section 3 include adding $\ell_2$-regularization to $w$ (Azizzadenesheli et al., 2019), thereby the loss becomes

$$\min_w \sum_{\hat{y}=1}^{k} \left( p_T(\hat{y}) - \sum_{y=1}^{k} p_S(\hat{y}, y) w_y \right)^2 + \xi \sum_{y=1}^{k} (w_y - w_y^{(0)})^2,$$

where $w^{(0)}$ is an initial IW estimate. One hyperparameter is introduced, $\xi$ (`lambda_iw`), which controls the strength of the regularization. When $\ell_2$-regularization is implemented with weight decay, the effective value of $\xi$ is multiplied by 2 times the learning rate of $w$ (`lr_iw`).

In our case, because IWDA is preceded by zero-shot fine-tuning whose decent performance means that the features are partially aligned, we initialize $w^{(0)}$ with the solution $w'$ to the unregularized IW loss on statistics obtained from the zero-shot model, computed using quadratic programming (Tachet des Combes et al., 2020). Since the initial estimate may be unreliable, we set $w^{(0)} = (1-\rho)w' + \rho \mathbf{1}$ with hyperparameter $\rho$ (`lambda_iw_init`).

Lastly, to minimize potential effects due to the miscalibration of neural models (Guo et al., 2017; Alexandari et al., 2020), hard confusion matrix and hard target output distribution are used in the estimation of IWs (Lipton et al., 2018; Garg et al., 2020).

## C.2 HYPERPARAMETER SETTINGS

We include the hyperparameter settings used in our experiments on mBERT, and the only difference in the settings on XLM-R is that the model learning rate is halved. The AdamW optimizer is used in all our experiments, and a weight decay of 0.01 is always applied. For a discussion on the methodology for hyperparameter tuning for unsupervised cross-lingual learning, see Artetxe et al. (2020c).

**Zero-Shot Fine-Tuning.** Learning rate is 1e-5 with 10% warmup and a linear schedule. Batch size is 8.

**IWDA.** Model learning rate is 1e-5 with 10% warmup and a linear schedule. Adversary learning rate is 5e-4 with a weight decay of 0.01, `lambda_gp` is 10, and `lambda_da` is 5e-3 with 10% warmup. `lr_iw` is 5e-4, `lambda_iw` (weight decay) is 2, and `lambda_iw_init` is 0.25. Batch size is 8 per domain (totals to 16 per step).

**Knowledge Distillation (KD).** Our implementation follows Hinton et al. (2015). Given a teacher LM fine-tuned either zero-shot or using IWDA, we distill its knowledge on a pre-trained student LM by minimizing the mean squared error (MSE) between student softmax outputs and teacher outputs, on source and target data without labels. If source labeled data is available, we also train the student to minimize source task loss.

We set the softmax temperature to 3, and the weight of the source task loss to 0.1. Learning rate is chosen from {5e-5, 1e-4, 2e-4}, and batch size is 32. As in (Wu et al., 2020), we freeze the bottom three layers of the student model during distillation.

**Self-Training (ST).** Our implementation follows Dong & de Melo (2019). Given an LM fine-tuned on source only, ST continues fine-tuning the model by iteratively assigning pseudo-labels to target unlabeled data and training on the ones with the highest confidence. Source loss is also minimized during self-training.

The hyperparameters of ST involve the number of pseudo-labeled examples to select and train on in each iteration. We set this to {0.05%, 2%} of the total number of unlabeled target examples. Learning rate is 1e-5, and batch size is 128.

**Number of Training Steps.** On sequence classification tasks, the number of training steps is equal to 4 epochs on source data. However, on token classification tasks (including NER), we do not set the number of steps according to the number of sequences in the dataset. This is because we noticed that the average length of the sequences differs between languages, which means

that the total amount of gradient updates received by each token may vary due to the averaging in the loss function: it is smaller on domains with longer sequence lengths. Therefore, we scale the number of steps on target language data using English data as the reference, so that the total amount of gradient updates received by each token is the same on average. The scaling is given by (average num. of tokens per sequence in target)/(average num. of tokens per sequence in English).

For example, on CoNLL, the scaling is 1.17 for German, 2.18 for Spanish, and 0.88 for Dutch, and fine-tuning for the equivalent of 4 epochs over the English portion with the batch size of 8 (7021 steps) means 7135 steps on German, 9086 on Spanish, and 6954 on Dutch.

