# OpenReview forum: "Cross-Lingual Transfer with Class-Weighted Language-Invariant Representations"
_ICLR.cc/2022/Conference — ICLR 2022 Poster_

### Official Review · Reviewer_WfZj · 2021-11-01

**Correctness:** 3
**Technical Novelty And Significance:** 4
**Empirical Novelty And Significance:** 4
**Recommendation:** 6
**Confidence:** 3

**Main Review:**

Strengths:

The proposed method is well motivated based the empirical analysis.

Experiments show effectiveness of the proposed method.


Weaknesses:

The experimental results are not convincing enough. 1) Only two of the tasks are considered. 2) only mBERT is tested. Would the same observations and conclusions hold when other widely-used pretrained models such as XLM are used. 3) According to Table 1, the proposed method is comparable with the existing work (Wu et al., 2020) on average. 4) There’s no baselines in Table 2. 5) Seemingly the proposed method is much more time consuming during training than prior works.


**Summary Of The Paper:**

This paper investigates a method for improving the cross-lingual transfer of pretrained multilingual models. The paper first empirically analyzed the influence of representation invariance and distributional class shift. Then, the paper proposed a method to improve the representation invariance and correcting the class shift. Experiments showed its superiority under large prior shifts.

**Summary Of The Review:**

The proposed method is well motivated given that the empirical analysis reveal the influence of representation invariance and class shift. Experiments show the effectiveness of the proposed method under large class shift. However, the experimental results are not convincing enough and the paper can be improved by conducting more experiments and analysis.

---

> ### Author Response · Authors · 2021-11-16
> **Response to Reviewer WfZj**
>
> We thank the reviewer for the comments and suggestions! We have added new results, and we hope our new results and responses have addressed the concerns below.
>
> ### Analyses and results on larger models and higher-level tasks
>
> In our revised version, we have added new analyses and experiment results with the XLM-R Large model (summarized in list of revisions).
>
> We also included new results on the XNLI dataset for checking whether the same observations in Section 2 hold on high-level tasks, and whether the principle of achieving domain alignment is also beneficial here. They are presented in Table 5 on page 18, with setup described in Appendix B on page 19.
>
> ### Additional baseline for MARC sentiment analysis experiments
>
> We would like to further emphasize here that our primary goal is not to claim new SOTA results on certain benchmark datasets by comparing with most of the existing approaches, but rather to design controlled experiments so that we could better explain the limitations of existing approaches as well as the benefits of the proposed conditional distribution alignment through importance weighting. Although not exhaustive, we hope the KD and ST baselines are representative of approaches based on semi-supervised learning.
>
> That being said, we still ran the method by Wu et al. (2020)—originally tested only on CoNLL NER with mBERT—on MARC and WikiANN NER (results included in Table 3 on page 8) but found very limited improvement compared to zero-shot.
>
> ### Significance of IWDA in light of more powerful methods and models
>
> The performance boost from IWDA under UCL settings will mainly come from enhanced alignment of features in the last layers. Without additional supervision or resources, IWDA is generally not capable of recognizing and learning entirely different hypotheses.
>
> However, we argue that the main strength of IWDA lies in its ability to detect prior shifts and improve on class-conditional feature alignment, which is a theoretical assumption for most SSL methods. By preceding concurrent UCL methods (and SOTA models in general) with IWDA as a preprocessing step, further performance gains could hope to be achieved and negative transfer under large prior shifts could be alleviated. We demonstrate this potential with results on IWDA + KD and IWDA + ST in (Tables 1, 2 and 3 on pages 7 and 8), as well as new results under prior shifts in Fig. 4 on page 9.
>
> ### Training time of IWDA
>
> The training time of IWDA is comparable to the time it would take to perform supervised training on both the source and target languages. While we restrict to four training epochs in our experiments, training for longer could lead to further gains since the adversary needs to learn a good representation of the data distribution in order to guide the alignment. This is illustrated with improved results on CoNLL (Table 5, page 18).
>
> In general, IWDA would take longer to optimize compared to conservative SSL methods that only train mildly, but the same amount of time as other alignment-based methods using adversarial training such as Keung et al. (2019). However, our method tackles the harder problem of both aligning representations and estimating prior shifts.

---

### Official Review · Reviewer_aVzf · 2021-11-03

**Correctness:** 4
**Technical Novelty And Significance:** 3
**Empirical Novelty And Significance:** 3
**Recommendation:** 6
**Confidence:** 4

**Main Review:**

Overall, the paper first presents insightful analysis that highlights the role of feature invariance and class prior shifts on the extent of zero-shot cross-lingual transfer. The insights from the analysis are then adapted to develop Importance-weighted Domain Adaptation for zero-shot crosslingual learning, resulting in improved performance on MARC sentiment analysis and WikiANN NER; with significantly improved robustness under class-prior shifts.

Strengths:
1. Well written and easy to understand.
2. Insightful analysis that grounds the presented approach.
3. Results from analysis with synthetically modified class prior distributions support the hypothesis that IWDA is improving robustness to class-prior shifts.

Weaknesses:
1. The paper evaluates on a limited set of tasks; having additional results on a wider range of tasks (for eg. additional tasks from the Xtreme benchmark) could significantly strengthen the results.

Suggestions / comments / questions:
1. Incorporating feature-invariance for domain adaptation has been studied for several NLP applications, including multilingual pre-training and zero-shot Neural Machine Translation. Discussion on several references is missing in the paper. [1, 2, 3]
2. Last paragraph on page 3: pratrained -> pre-trained
3. While the analysis in Figure 1 suggests that F1-score is directly correlated with conditional feature shift, is it possible there are other confounds in this analysis; for eg. the amount of pre-training data used for each of these languages?
4. Could the relatively weaker results on NER also be caused by challenges with aligning representations at a sub-word level across languages (with different levels of tokenization granularity across different languages)?

References:
[1] Explicit Alignment Objectives for Multilingual Bidirectional Encoders, Hu et al.
[2] The Missing Ingredient in Zero-Shot Neural Machine Translation, Arivazhagan et al.
[3] Improving Zero-shot Translation with Language-Independent Constraints, Pham et al.

**Summary Of The Paper:**

The paper first demonstrates the importance of feature invariance (language invariant representations) and class-prior invariance across languages on zero-shot cross-lingual performance. By analyzing the zero-shot performance on different languages on the MARC reviews and WikiANN NER tasks, and comparing against the class conditional distance between the source language (English) and target language feature representations, the authors illustrate the how high feature invariance results in better zero-shot performance. By also synthetically modifying the class prior for the target language, the authors demonstrate how increasing differences in class priors result in decreasing zero-shot cross-lingual transfer performance.

Building on their observations, the authors propose an approach that:
(i) Introduces an adversarial loss term to penalize distortion in average class conditional feature representations between the source and target languages.
(ii) Adds an importance weighting term to ensure the approach doesn't fail under class prior shifts.

Empirical studies demonstrate that the proposed approach improves significantly over the vanilla zero-shot model on both MARC sentiment analysis and NER tasks, also improving over self-training on sentiment analysis. Comparisons on synthetic datasets (sub-sampled from NER and MARC datasets) that enhance the class prior shift highlight the robustness of the approach under large class prior shifts where previous approaches fail.

**Summary Of The Review:**

Overall, the paper first presents insightful analysis that highlights the role of feature invariance and class prior shifts on the extent of zero-shot cross-lingual transfer. The insights from the analysis are then adapted to develop Importance-weighted Domain Adaptation for zero-shot crosslingual learning, resulting in improved performance on MARC sentiment analysis and WikiANN NER; with significantly improved robustness under class-prior shifts.

The paper produces some valuable insights and develops a well-grounded approach that improves the robustness of zero-shot crosslingual learning. However, the empirical results are limited to just two (relatively) small scale tasks. Having additional results on a wider range of tasks (for eg. additional tasks from the Xtreme benchmark) could significantly strengthen the results.

Given the thorough analysis, but limited range of tasks, I am leaning towards acceptance. If authors include additional empirical results I would be willing to update my recommendation to strong accept.

---

> ### Author Response · Authors · 2021-11-16
> **Response to Reviewer aVzf**
>
> We thank the reviewer for the comments and the support! We are grateful for the comment that the paper is well-written, and we have further revisited and significantly refined the presentation of the results and the algorithm. We also included the references from machine translation literature and made stylistic changes to the paper following the suggestions. We hope that the revised paper now has more of what the reviewer liked, not less.
>
> ### Additional analysis and results on higher-level tasks
>
> While the main goal of our experiments is to illustrate limitations on existing approaches under prior shifts, following the reviewer’s suggestions, we included new results on XNLI for studying whether our principle of achieving domain alignment is beneficial to high-level tasks. They are presented in Table 5 on page 18, with setup described in Appendix B on page 19.
>
> On XNLI, we also produced and included a plot, similar to the one in Section 2.1, that compares the performance to representation alignment, and found that the same findings hold on this higher-level task (Fig. 7, page 16).
>
> ### Potential confounders in analysis
>
> In an effort to establish that the relations we found between performance, representation alignment and prior shift hold regardless of linguistic properties and pre-training resourcefulness (meaning that these factors should only affect the learned representation but not its relation to performance), in Fig. 6 on page 16, we included the same plots in Section 2 but produced on WikiANN NER dataset. Specifically, we evaluated 39 languages of varying degrees of linguistic similarity to English and magnitudes of pre-training corpus size. We kindly refer to Table 5 on page 13 of Hu et al. (2020) for a description of these languages, and Table 1 on page 4 of Wu and Dredze (2020) for their pre-training corpous sizes in mBERT.
>
> ### Effects of tokenization
>
> Following the observation from word embedding literature that embeddings from different languages can be orthogonally aligned, seemly due to having similar word-level co-occurrence statistics, we speculate that potential caveats with subword vocabulary could be avoided, if the model has the ability to “group” subwords belonging to the same word together in their feature representations during contextualization. While the results from Libovický et al. (2020) suggest that mBERT possesses this ability, we noticed that later versions of BERT added “whole word masking”, which could explicitly guide the model to treat subwords as the same entity. Since it was not applied in their latest mBERT release, it would be interesting to see whether further improvements could come from it.
>
> A related remark is that if the tokenization scheme is different in pre-training and fine-tuning, then no transfer of knowledge may take place. For example, WikiANN Thai (th) data is tokenized differently from mBERT (the former splits Thai characters with diacritical marks into more than one token), and it sees near-zero performance under zero-shot (Hu et al., 2020). On the other hand, mBERT transfers to Thai on XNLI (Wu et al., 2019).
>
> ### References
>
> - Hu et al. (2020). XTREME: A Massively Multilingual Multi-task Benchmark for Evaluating Cross-lingual Generalization. https://arxiv.org/pdf/2003.11080.pdf#page=13
> - Wu and Dredze (2020). Are All Languages Created Equal in Multilingual BERT? https://arxiv.org/pdf/2005.09093.pdf#page=4
> - Libovický et al. (2020). On the Language Neutrality of Pre-trained Multilingual Representations. https://arxiv.org/pdf/2004.05160.pdf
> - Wu et al. (2019). Beto, Bentz, Becas: The Surprising Cross-Lingual Effectiveness of BERT. https://arxiv.org/pdf/1904.09077.pdf

---

### Official Review · Reviewer_nGDq · 2021-11-03

**Correctness:** 2
**Technical Novelty And Significance:** 2
**Empirical Novelty And Significance:** 2
**Recommendation:** 6
**Confidence:** 3

**Main Review:**

The paper provides extensive analyses on cross-lingual transfer in the commonly used approaches that follow the pretrain-fine-tune strategy. In this strategy, the multilingual model finetuned on English task data is known to have zero-shot capability in the other languages, however, it is not well studied that, in unsupervised cross-lingual learning such as multilingual language model, what the role of shared representations is. Considering this, the paper gives a good start with substantial analyses that are helpful to understand what are the key factors to successful cross-lingual transfer learning.
Since XTREME benchmark provides a variety of multilingual NLP tasks, the experiment section could be extended with more results. The targeted languages in the current experiment are considered as a high-resource language. It would be better if the authors could move the remaining results into the main 0 pages and give more discussion on them.

**Summary Of The Paper:**

In this paper, the authors provide substantial analyses on the cross-lingual transfer performance in the multilingual neural language models and reported that the performance is strongly correlated with representation invariance and negatively affected by distributional shift in class priors between data in the src/tgt languages. Based on these findings, the authors propose an unsupervised cross-lingual learning method, called importance-weighted domain adaptation (IWDA), where it performs feature alignment, prior shift estimation, and correction. The authors experimented on two different NLP tasks such as multilingual NER and multilingual sentiment analysis tasks, and experimentally showed the effectiveness. Besides that, they demonstrated that the proposed approach improves performance further, when combined with existing semi-supervised learning approaches.

**Summary Of The Review:**

The paper is mostly well organized, and provides extensive analyses and experimental results. They will be helpful to the studies in crosslingual transfer learning. However, some important descriptions on the model settings or experimental results are shown in Appendix, which makes the paper difficult to read. I would suggest to revisit and reorganize the sections for better readability. Regarding the experiments, since mBERT that the authors used as a pretrained model serves diverse language representations, they could provide deeper discussion, by moving Table 4 to the main 9 pages.
####
I read the responses from authors, and appreciate their response and showing more results. I am okay with accepting the paper, but keep the score 6. Because one concern might be that those results/analyses are mostly described in Appendix. The authors would need to reconstruct the manuscript by moving them into the main pages.

---

> ### Author Response · Authors · 2021-11-16
> **Response to Reviewer nGDq**
>
> We thank the reviewer for the suggestions! We revamped the appendix for improved readability, and improved the presentation of the IWDA algorithm, where Section 3 is refined and kept to a high-level description with discussions on the connection to findings in Section 2. We also plan on releasing our Hugging Face Transformers-compatible code on a public repository.
>
> ### Results on higher-level tasks and low-resource languages
>
> While the main goal of our experiments is to illustrate limitations on existing approaches under prior shifts, following the reviewer’s suggestions, we included new results on XNLI for studying whether our principle of achieving domain alignment is beneficial to high-level tasks. They are presented in Table 5 on page 18, with setup described in Appendix B on page 19.
>
> Besides new results on XNLI low-resource languages, following the reviewer's suggestion, we also highlight results on WikiANN low-resource languages in Table 2 on page 7, previously buried in the appendix (Bengali, Burmese and Urdu, also considered in Bari et al. (2021)).

---

### Official Review · Reviewer_gtqu · 2021-11-04

**Correctness:** 3
**Technical Novelty And Significance:** 2
**Empirical Novelty And Significance:** 3
**Recommendation:** 6
**Confidence:** 5

**Main Review:**

===== UPDATE AFTER THE RESPONSE =====
I would like to thank the authors for the new set of experiments and their very thorough author response. The new experiments indeed strengthen the paper and better outline the key contributions of the work. Some of my main concerns still do remain though: I am not fully certain that, given its mixed performance in the transfer tasks, the proposed framework will be very useful with more powerful transfer methods. This preliminary evidence has not fully convinced me on its future impact, and I am still not fully convinced by the choice of baselines, and whether to understand this preliminary empirical evidence as fertile ground for future enhancements in UCL. Some other clarifications were provided in the response, so I have increased my score, but as said - I still have reservations when it comes to the paper's empirical contributions and consequent impact.
=====


Overall, the paper provides some nice insights, especially treating the problem of cross-lingual transfer as largely a domain transfer problem, which allows it to use some domain adaptation (DA) machinery from prior work and apply it to the UCL problem. The empirical gains, although not huge, do demonstrate the alignment between the research hypothesis and empirical scores. The main strength of the paper imo is Section 2 which delves deeper into analysing factors that affect cross-lingual transfer.

The ideas of having more invariant representations to improve cross-lingual transfer are not new, and they date back to the work on cross-lingual word embeddings (e.g., see the work of Dubossarsky et al. EMNLP 2020; or Patra et al., ICLR 2019, or Zheng et al., ACL 2019). Also the idea of treating cross-lingual transfer as a DA problem is also not novel - e.g., PhD thesis of Ruder discusses the similarity of the two problems in a nice and detailed way.

One of the major concerns I have with the current paper and its current presentation is that it is very difficult to discern between novelty of this work and what was done in prior research and simply reapplied to UCL. For instance, it seems a bit that this work is a (largely incremental) application of the prior (more fundamental) work of Tachet des Combes et al. (NeurIPS 2020) to a new (but highly similar) problem. For instance, the authors should clearly note in Section 3 if they bring any methodological contribution here or they just describe the previous method of Tachet des Combes et al.

In a similar vein, the paper can also be seen as an extension of the work from Keung et al., adding this mitigation of prior class shift into the mix, which yields slight improvements.

I am also not convinced by the results and the entire evaluation protocol, detecting several potential problems and weaknesses here:
- Evaluation is conducted only on high-resource languages (from the NLP perspective), while the main promise of UCL is to improve on NLP tasks for lower-resource languages as done in plenty of contemporary NLP research. I wonder how IWDA would behave for such languages and whether it would bring any benefits. Doing zero-shot transfer from EN to DE really does not make much sense imo...
- Evaluation is conducted on two reasonably simple tasks: NER and MARC sentiment analysis (which is a classification task with a small number of classes) - a more detailed empirical analysis on other higher-level semantic tasks (e.g., XNLI, PAWS-X) is warranted for a clearer picture of the benefits of the proposed approach.
- The gains are typically not very high, and are close to none (when compared to the simple alignment method of Wu et al.) in the NER task. Moreover, performance on MARC sentiment is measured only against the vanilla zero-shot transfer and an older self-training (ST) baseline. What precludes the authors to again compare against Wu et al. or some other more recent techniques that go beyond the simplest zero-shot transfer protocol?
- The paper does not really put results into the perspective of plenty of related work that has been conducted in this area recently: e.g., check the leaderboards of XTREME and XGLUE for more sophisticated baselines: a true gain would be showing that applying IWDA along with these stronger models can yield further benefits. With the current set of experiments, I believe that the paper will have very limited impact.
- Along the same line, why is only mBERT evaluated? All concurrent work also provides evaluations with a stronger XLM-R Base model, and I also wonder whether these gains would remain with a larger and an even stronger XLM-R Large model.

The paper also misses some very relevant related work, e.g., it seems very close to this work in its optics and design: https://arxiv.org/pdf/2011.11499.pdf
- I would like to see a discussion regarding their (dis)similarity.
Some other papers that should have been briefly discussed and cited in this paper:
- https://arxiv.org/pdf/2104.07908.pdf
- https://arxiv.org/pdf/2005.00396.pdf (analysis of mBERT's multilinguality)

**Summary Of The Paper:**

This paper looks at the problem of unsupervised cross-lingual transfer (termed UCL) in the paper through the optics of domain adaptation. After empirically analysing and validating that distributional shifts in class priors might cause a huge problem for UCL (which wasn't tackled in previous research), the authors proceed with an introduction of a new method that aims to mitigate that problem. The idea is to get rid of that shift through a approach called importance-weighted domain adaptation (IWDA), which is largely the adaptation of the work from Tachet des Combes et al. (NeurIPS 2020) to the UCL problem.

The results on two tasks in the UCL setup (NER and MARC classification) show slight gains over the standard zero-shot transfer when IWDA is applied, with more prominent gains reported when a stronger domain shift is observed - however, such a setup has been created mostly artificially, to further demonstrate the benefits of modelling the shift in the model.

**Summary Of The Review:**

The paper presents a DA-inspired view on (unsupervised) cross-lingual transfer, offering some insightful analyses, but it seems as an eclectic (mostly incremental) work, with insufficient and lacking empirical validations, incomplete baselines, and inadequate positioning against previous work in this area, which would negatively affect its potential impact and its overall contributions - more work and a stronger empirical foundation are needed.

---

> ### Author Response · Authors · 2021-11-16
> **Response to Reviewer gtqu**
>
> We thank the reviewer for the thorough comments! We have included the references, and added results on XLM-R Large (summarized in the list of revisions). We hope that our update has addressed most of the concerns, and we are happy to answer any further comments.
>
> ### Methodological contributions, and comparison to Keung et al. (2019)
>
> In our revised Section 3, we now highlight our improvements to the stability of the IW estimator based on issues that were not discovered in Tachet des Combes et al. (2020). These changes are essential for good performance on NLP tasks with long training episodes.
>
> Algorithmically, IWDA simply adds an IW estimation module into the domain alignment (DA) technique by Keung et al. (2019). From a theoretical perspective, these two objectives aim for very different goals. IWDA hopes to achieve *importance-weighted* representation alignment—a necessary condition for the ideal goal for *class-conditional* alignment. The DA objective optimizes for *unweighted* alignment, which has a lower bound on performance under prior shifts and could result in negative transfer, as we show at the end of Section 2.2 (page 5). In fact, this is precisely our motivation in using IWDA to improve over the existing DA approaches.
>
> ### Results on higher-level tasks and low-resource languages
>
> While the main goal of our experiments is to illustrate limitations on existing approaches under prior shifts, following the reviewer’s suggestions, we included new results on XNLI for studying whether our principle of achieving domain alignment is beneficial to high-level tasks. They are presented in Table 5 on page 18, with setup described in Appendix B on page 19.
>
> Besides new results on XNLI low-resource languages, we also highlight results on WikiANN low-resource languages in Table 2 on page 7, previously buried in the appendix (Bengali, Burmese and Urdu, also considered in Bari et al. (2021)).
>
> ### Additional baseline for MARC sentiment analysis experiments
>
> We would like to further emphasize here that our primary goal is not to claim new SOTA results on certain benchmark datasets by comparing with most of the existing approaches, but rather to design controlled experiments so that we could better explain the limitations of existing approaches as well as the benefits of the proposed conditional distribution alignment through importance weighting. Although not exhaustive, we hope the KD and ST baselines are representative of approaches based on semi-supervised learning.
>
> That being said, we still ran the method by Wu et al. (2020)—originally tested only on CoNLL NER with mBERT—on MARC and WikiANN NER (results included in Table 3 on page 8) but found very limited improvement compared to zero-shot.
>
> ### Significance of IWDA in light of more powerful methods
>
> The performance boost from IWDA under UCL settings will mainly come from enhanced alignment of features in the last layers. Without additional supervision or resources, IWDA is generally not capable of recognizing and learning entirely different hypotheses.
>
> However, we argue that the main strength of IWDA lies in its ability to detect prior shifts and improve on class-conditional feature alignment, which is a theoretical assumption for most SSL methods. By preceding concurrent UCL methods (and SOTA models in general) with IWDA as a preprocessing step, further performance gains could hope to be achieved and negative transfer under large prior shifts could be alleviated. We demonstrate this potential with results on IWDA + KD and IWDA + ST in (Tables 1, 2 and 3 on pages 7 and 8), as well as new results under prior shifts in Fig. 4 on page 9.
>
> ### Discussions on Li et al. (2020)
>
> In addition to cross-lingual transfer, this paper also considers the problem of domain shift, and proposes a zero-shot approach based on the same principle of learning invariant representations. By using pre-trained XLM models trained with MLM and TLM as language-invariant feature extractors, they train two feature extractors on top of XLM outputs (while freezing XLM) to extract domain-invariant and domain-specific features via minimizing MI between the two, and maximizing MI between the domain-invariant head and XLM outputs.
>
> Since TLM requires bitext for pre-training, this limits their approach to high-resource languages, whereas our method attempts to learn invariant representations without cross-lingual supervision. However, it would be interesting to investigate whether UCL can benefit from their approach for extracting *domain*-invariant features, which is motivated from an information theoretical perspective. IWDA can also be viewed from this perspective, in that the alignment objective is aimed at minimizing MI between features and language/domain identity, and the source loss term is added to maximize MI between inputs and features.

---

### Author Response · Authors · 2021-11-16
**List of Revisions**

We would like to thank all the reviewers for the feedback. We have improved our presentation and included additional analysis and experiment results.

A summary of the new results is as follows:

- Included the method by Wu et al. (2020) as a baseline for MARC (Table 3, page 8).
- Added IWDA + KD and ST results under prior shifts on CoNLL and MARC (Fig. 4, page 9).
- Added experiment results on XNLI textual entailment task (Table 5, page 18), with setup described in Appendix B on page 19.
- Plot of performance vs. alignment on XNLI dataset (Fig. 7, page 16).
- Plot of performance vs. alignment and prior shift on MARC with XLM-R Large (Fig. 8, page 17).

We have also added results of the XLM-R Large model on most datasets. Due to significant training time, we prioritized IWDA experiments:

- CoNLL NER (Table 1, page 7)
- WikiANN NER (Table 2, page 7)
- MARC sentiment analysis (Table 3, page 8)
- MARC prior shift experiments (Fig. 10, page 19)
- XNLI textual entailment (Table 5, page 18)

Finally, the improvements on the presentation include:

- Restructured the description of the IWDA algorithm to highlight our technical contributions. Section 3 is now kept to a high-level description with discussions on the connection to findings in Section 2.
- Revised Section 4, incorporating new results and clarifying the discussion points.
- Revamped the appendix for better readability that details additional results.
- To allow readers to get an immediate idea of our approach, we tweaked the full name of IWDA to importance-weighted domain *alignment*.
- Made stylistic changes and included missing references and discussions as suggested by the reviewers.

We hope our new results and responses have successfully addressed most of the reviewers' concerns, and we are happy to answer any further questions if needed. We look forward to your updated reviews.

---

### Decision · Program_Chairs · 2022-01-20

**Decision:**

Accept (Poster)

**Comment:**

The paper presents a domain adaptation approach based on the importance weighting for unsupervised cross-lingual learning. The paper first analyzes factors that affect cross-lingual transfer and finds that the cross-lingual transfer performance is strongly correlated with feature representation alignments as well as the distributional shift in class priors between the source and the target. Then the paper designs an approach based on the observations.

Pros:
+ The paper is well written and the proposed approach is well motivated.
+ The analysis about which factors affect cross-lingual transfer is interesting and provides some great insight.

Cons:
- As the reviewer pointed out, the experiments for verifying the proposed approach are relatively weak.

Overall, the paper presents nice insights to connect cross-lingual transfer with domain adaptation. All reviewers lean to accept the paper and I also found the paper is in general interesting.